# Healthy-Canteen Displays: A Tactic to Encourage Community Sport Canteens to Provide Healthier Food and Beverage Options

**DOI:** 10.3390/ijerph181910194

**Published:** 2021-09-28

**Authors:** Gina Trakman, Kiera Staley, Adrienne Forsyth, Brooke Devlin, Anne Skiadopoulos, Karen Pearce, Matthew Nicholson, Regina Belski

**Affiliations:** 1School of Allied Health, Human Services and Sport, La Trobe University, Melbourne 3086, Australia; a.forsyth@latrobe.edu.au (A.F.); b.devlin@latrobe.edu.au (B.D.); 2Centre for Sport and Social Impact, La Trobe University, Melbourne 3086, Australia; k.staley@latrobe.edu.au (K.S.); m.nicholson@latrobe.edu.au (M.N.); rbelski@swin.edu.au (R.B.); 3Learning and Teaching, RMIT University, Melbourne 3001, Australia; anne.skiadopoulos@gmail.com; 4Basketball Victoria, Melbourne 3152, Australia; karen.pearce@footballvictoria.com.au; 5Football Victoria, Melbourne 3004, Australia; 6Research and Development, Monash University Malaysia, Subang Jaya 47500, Malaysia; 7School of Health Sciences, Swinburne University of Technology, Melbourne 3122, Australia

**Keywords:** front of package label, sugar sweetened beverages, nudge, cafeteria, canteen, sport

## Abstract

(1) Background: Community sport settings present a range of conflicting health behaviours, including the tension between being physically active and consuming discretionary foods. Therefore, community sport settings are considered a promising location for health promotion. The aim of this project was to evaluate perceptions, knowledge and the impact (e.g., barriers and outcomes) of a healthy-canteen (cafeteria) display, based on traffic light labeling (TLL), which was set up at an Australian Basketball Association Managers’ Convention and Trade Show. (2) Methods: We set up a healthy ‘canteen display and surveyed Basketball managers on their perceptions of the display before (Survey 1) and after (Survey 2) visiting the display. Three months later they were surveyed (Survey 3) on changes made to their community sport canteens. (3) Results: Eighty-eight, 76 and 22 participants completed Surveys 1, 2 and 3, respectively. Participants believed stocking healthy foods and beverages was important (mean 8.5/10). Food waste, lack of consumer interest and price were identified barriers to stocking healthy foods. After visiting the display, 75% were inspired to make changes and 50% were surprised by the differences between their perceptions of the healthfulness of foods and the TLL ratings. Post-convention, 41% and 70% made or had planned healthy changes to their community sport canteen. (4) Conclusions: A healthy-canteen display is a low-cost, easy-to-implement strategy that may be able to direct self-driven improvement in the healthfulness of foods stocked at community canteens and lead to improved nutritional intakes at these venues.

## 1. Introduction

More than 25% of Australians play organised sport, with 10% of the population involved in non-playing roles [1]. Individuals benefit from engaging with sport but it is unclear whether being physically active clusters with other health-promoting behaviours [2]. Some studies have found a positive relationship between superior diet quality and physical activity [3,4], but sport participation has also been associated with increased alcohol use [5]. Alcohol and energy-dense, nutrient-poor, ultra-processed foods (i.e., ‘discretionary’/‘junk’ foods, such as cakes, muffins, pastries, biscuits and soft drinks) are often promoted and advertised via sport [6,7,8] and junk food companies frequently sponsor sport [9]. Likewise, national and community sporting venues often supply patrons with discretionary foods, with investigators even categorising certain recreational sporting facilities in America as ‘obesogenic’ environments [10]. Food at community sporting venues is usually supplied via community sport canteens, which are also referred to as stores, kiosks, concession stands or cafeterias. Community sport canteens sell (predominately) readymade and quick-serve food and drink, and may be run by volunteers.

The community sport setting presents a range of conflicting health behaviours, foremost among them the tension between being physically active and consuming discretionary foods, including sugar sweetened beverages (SSBs) [10,11,12]. As such, community sport canteens are considered a promising location for health promotion [13]. Sport venues fit a ‘settings-based’ approach to health promotion, which acknowledges that the settings in which we enact daily activities significantly influence our health [14]. Moreover, where sport venues receive funding from state government, it is reasonable to expect they engage in health promotion [15], as recommended by the World Health Organisation (WHO) [16].

A common and successful settings-based strategy to promote healthy foods in sporting venues is traffic light labelling (TLL) [17]. TLL classifies foods as green (healthy/best choices, e.g., water), amber (choose carefully, e.g., muesli bars), and red (limit, e.g., chocolate) based on their saturated fat, added sugar, and/or salt, energy and fibre content [18]. TLL has been shown to increase sales of ‘green’ options in a community sport canteen in Canada without reducing revenue [19]. Likewise, TLL has successfully reduced energy and saturated fat provided to customers at a community sport canteen in Australia [20]. Several other examples of successful health promotion activities within Australian community sport canteens exist, including: removing SSBs from community sport canteens [21], coaches providing healthy snacks to children [22], and a multi-faceted approach focused on development of food and nutrition policies, increasing availability of fruits and vegetables, as well as promoting healthy foods using posters, meal deals, competitive pricing strategies, and the education of players [23].

Although there is evidence that health promotion through community sport canteens can work in controlled studies, most community sport clubs do not actually have healthy eating policies in place [24]. Moreover, even when policies on the promotion of healthy food and drink exist, they are often not implemented due to lack of strategies regarding policy enactment [25], in addition to other barriers [26] such as concerns about training canteen staff, who are often volunteers that turnover on a frequent basis [27], and about food waste and lack of consumer interest, which could affect the ability of the canteen to make a profit [28,29]. Profit-making is important as sport canteens often serve as an essential revenue source for the sporting venue or club [21], while many organisations receive sponsorship from ‘junk food’ companies, and believe a rejection of these could lead to significant financial strain [8].

Common factors supporting the implementation of healthy-canteens at sporting venues include organisational capacity building [30], support from health promotion bodies [25], and importantly, the nutrition knowledge and perceptions of venue managers [31]. Despite the importance of venue managers in influencing canteen policies and practices, little is known about effective tactics to impact their perceptions, encourage them to adopt policies designed by health promotion bodies or influence them to engage in self-directed change. Self-directed change is likely to be a low-cost tactic with potential impact across the sport industry. Accordingly, the aim of this project was to evaluate perceptions, knowledge and the impact (e.g., barriers and outcomes) of a healthy-canteen display, based on traffic light labelling (TLL), that was set up at a Basketball Association Managers’ Convention and Trade Show. We hypothesised that the intervention would positively impact managers’ perceptions of healthy-canteens and their ability to introduce healthier items in their local contexts. This project was supported by the Victorian Health Promotion Foundation (VicHealth), the broad aims of which are to create healthier sport environments, increase access to drinking water and make healthier food and beverage choices the easy choice in a range of environments [32].

## 2. Materials and Methods

### 2.1. Study Design

This quasi-experimental, pre-test post-test study involved the set-up and evaluation of a display healthy-canteen.

### 2.2. Setting

The display canteen was set-up at a Basketball Association Managers’ Convention and Trade show, 9–11 March 2017, Melbourne, Victoria, Australia. The convention focused on “facility management, programs and activities, technology, human resources and high performance” [33].

### 2.3. Particpants

All delegates aged 18 years and over who attended the convention were invited to participate (convenience sample) and no exclusion criteria were applied. A total of 164 associations were invited to attend but the actual number of attendees on the day was not recorded; 88 delegates participated in the current project.

### 2.4. Healthy-Canteen Set-Up

The healthy-canteen display used TLL to classify foods as green (best choices), amber (choose carefully) or red (limit), and demonstrated examples of healthy (green) products and how to present them. To challenge perceptions that healthy food can increase food waste [28], several ideas for non-perishable ‘green’ items were displayed (e.g., popcorn) (Figure 1). Green, amber, and red classifications were based on pre-defined criteria for saturated fat, added sugar, and/or salt, energy and fiber quantity [18]. Examples of green, amber, red foods include fresh fruit, muesli bars, and chocolate respectively. For further details on how we classified items, we direct readers to [34,35,36].

### 2.5. Healthy Canteen Evaluation

The impact of the display canteen on managers’ perceptions of healthy-canteens and TLL was assessed via a short questionnaire, administered using an iPad, before delegates visited the stand (Survey 1: baseline survey) and immediately after they visited the stand (Survey 2: evaluation survey). All participants were approached on entry to the convention to complete Survey 1 and all participants who visited the display were invited to complete Survey 2. After completing Survey 2, we gave respondents a gift bag with samples of healthy packaged food, lists of suppliers, costing ideas, and suggestions on how to make changes to their canteens. The potential of the display and gift bags to encourage managers to stock healthier foods was assessed via short survey (Survey 3: follow-up survey), administered via email to all participants who had completed Survey 2, three months after the convention.

### 2.6. Survey Details

Variables and data sources: All surveys (1–3) asked who ran and operated the club canteen (no canteen/private/club/council/other) and if respondents could influence products stocked at the canteen (yes/no). Surveys 1, 2, and 3 had an additional nine, eight and five questions, respectively. Response options for Surveys 1 and 2 were yes/no, Likert-scales (1–10) or open-ended. Response options for Survey 3 were yes/no or multiple choice.

Survey 1 (baseline survey): Two questions assessed awareness of, and confidence in implementing the traffic light system. The remaining questions assessed perceived: healthfulness of current foods stocked at the canteen; importance of having healthy canteen options; barriers to implementing a healthy canteen; impact of healthy canteens on sales; and utility of seeing a display healthy canteen. The open-ended questions probed respondents about currently available healthy canteen options and what they thought a healthy canteen would look like. Authors derived barriers to implementing a healthy canteen based on existing literature; respondents were instructed to choose all options that applied [29].

Survey 2 (evaluation survey): Two questions on perceptions of importance of including healthy foods and confidence in using the traffic light system were directly repeated from Survey 1. The remaining four questions assessed how useful delegates found the canteen, whether the display provided ideas for products to stock, if the traffic light system was surprising, and whether the canteen inspired delegates to make any changes to their canteen. The open-ended questions asked: what ideas respondents liked, what surprised them with regards to traffic light classifications, what other information they would find helpful, and what types of changes they would consider making.

Survey 3 (follow-up survey): The follow-up survey checked if respondents visited the healthy canteen display at the Basketball Managers Convention and Trade Show. The remaining items asked if any changes had been made and what these were, or if any changes were planned and what these were. Respondents were instructed to choose all options that applied. Due to time constraints, the surveys were not validated.

All surveys are provided as Appendix A. 

### 2.7. Statistical Methods

No *a priori* power calculation was performed. Frequency analysis of responses to all questions across surveys were calculated. For yes/no and multiple-choice questions, percentage of each response was calculated. For Likert-scale questions, frequency of response and mean response was calculated. Mean change in perceived importance of having healthy options at a canteen and confidence in using the traffic light system before and after visiting the display stand was assessed using an independent *t*-test (equal variances assumed). A summary of written responses to open ended questions were described in text. All surveys that were started were completed, thus there was no need to handle missing data.

### 2.8. Ethical Considerations

The front page of each online survey included background information about the research, along with participant ethical considerations should they choose to complete it. Given the low-risk nature of the research, consent to take part was implied through the lack of objection to complete the relevant survey (see Institutional Review Board Statement for ethics approval details).

## 3. Results

### 3.1. Participants

88, 76 and 22 individuals completed Survey 1 (baseline), Survey 2 (evaluation) and Survey 3 (follow-up), respectively. In order to ensure anonymity, and in line with our planned data analyses, sociodemographic data was not collected. General response rates could not be calculated because the numbers of attendees at the convention was not recorded. Of the 164 associations who were invited to attend the convention, 32 (20%), 25 (15%) and 16 (9%) were represented in each survey. In addition, 8, 7 and 3 interstate or New Zealand associations were represented. Fifty-percent of associations that completed the baseline survey were represented in the follow-up survey, which is reflective of expected survey response rates [37].

### 3.2. Operation of Canteens and Ability of Respondents to Influence Options

About half the canteens (47–55%) were club owned and operated. In Survey 1 (baseline survey) only 34% of respondents reported that they could influence which products were available. This increased to 39% in Survey 2 (evaluation survey) and 68% in Survey 3 (follow-up survey). This is likely reflective of individuals with decision-making power being more interested in continuing participation in the research. Respondents whose club had a canteen were also more likely to complete Surveys 2 and 3 (Table 1).

### 3.3. Survey 1 (Baseline Survey)

The mean ratings for importance of having healthy foods at a canteen was 8.5 out of 10. Most respondents (92%) thought seeing a display of a healthy-canteen would be useful. While just over half (52%) the respondents had heard of TLL, 91% rated their confidence in being able to classify foods based on the TLL as 5 or above (out of 10), with a mean confidence rating of 7.4. Nearly half (45%) the respondents thought having more healthy foods would have no impact on sales (Table 2).

The most frequently selected barriers to having healthy foods available were increased food waste (40%), people not wanting healthy foods (34%), price (29%) and negative impact on sales (27%) (Figure 2). Open-ended responses to healthy options currently stocked at canteens included: water, fruit, sandwiches/salad rolls, smoothies, muesli bars and protein bars. Open-ended responses regarding what a healthy canteen would look similar to included: ‘lower sugar items’, ‘low carb high fat items’, ‘fruit and salad options’, ‘get rid of fries’, ‘less processed foods’, ‘appealing’ and ‘fresh displays’.

### 3.4. Survey 2 (Evaluation Survey)

Viewing the display did not lead to a statistically significant change in mean rating for importance of stocking healthy options, *t* (−0.825), df = 162, *p* = 0.410. Likewise, there was no statistically significant change in confidence in classifying foods according to TLL between Survey 1 and Survey 2, *t*(−0.558), df = 162, *p* = 0.578 (Table 3).

Mean rating for usefulness of the healthy-canteen display was 8.0 out of 10. 92% of respondents reported the healthy-canteen display had provided them with new ideas about foods to stock at their canteen and 75% reported they were inspired to make changes to their canteens. 50% stated they found the colour coding system for some products surprising in relation to their expectations (Table 3).

Open-ended responses indicated unexpected ratings for: pre-boxed salads, savoury muffins, bars/muesli bars, sport drinks, juice and low-fat fortified milk beverages. One manager stated they thought ‘chia bars’ would be red and not amber, and another thought liquid breakfast drinks would be amber and not green; however, for other responses it was not possible to tell if foods were more or less healthy than expected. The ideas that managers liked included: wraps, salads, snack packs, fruit and vegetable sticks. Participants also reported that being provided with information on the physical layout and appearance of the canteen display would be helpful for them to make changes, noting the impact of ‘decoration’ and ‘bright packaging selling better’. Many responses also indicated more information was desired (‘more information’; ‘more options and ideas’; ‘more food variety’; ‘brochures’; ‘flyers on the food options’).

### 3.5. Survey 3 (Follow-Up Survey)

All respondents (n = 22) who completed the follow-up survey had visited the healthy-canteen display and 95% had a canteen at their venue. 41% (n = 9) had made changes to their club canteens as a result of visiting the display and 70% (n = 14) reported having plans to make changes. The most common change already made (67%; n = 6) and planned (71%; n = 10) was ‘product supply and availability’. Changing product visibility [44% (n = 4) already made; 50% (n = 7) planned] and ‘marketing and promoting’ [22% (n = 2) already made; 57% (n = 8) planned] were also popular choices (Figure 3).

There were only two open-ended responses provided by participants, which were ‘talk with the owners about new healthy options’ for already made changes and ‘more healthy options during events’ for planned changes. For additional resources, respondents requested details of food suppliers, signage options, ideas for working with the local council, photo displays of other canteens and advice on storage of fresh food/disposal of waste/how much waste to expect. Six associations requested additional support from Basketball Victoria, the governing body in the State.

## 4. Discussion

There is evidence that TLL can lead to more healthful food and beverage choices within sporting [19], school [38,39] and other point-of-purchase environments [17]. There is also evidence of success for interventions focused on additional provision and promotion of fruit, vegetables and non-SSBs, especially when these are implemented with human resources and workforce development support [23]. However, to our knowledge, this is one of the only studies to directly evaluate the effectiveness of a tactic aimed at encouraging sporting venues to implement healthier canteens in a self-directed manner. The key findings were: (1) basketball managers, such as presidents in related sports settings [28], believed stocking healthy food was important but reported barriers to having a healthy-canteen; (2) over 90% of basketball managers found the healthy-canteen display helpful; (3) most attendees were inspired to make changes to their own canteens after viewing the healthy-canteen display; and (4) over 40% of respondents to the final follow-up survey reported making changes to their canteen.

Taken together, our results indicate that a healthy-canteen display may be able to convince sport venue managers to adopt healthier canteens. This is consistent with existing literature showing that organisational capacity building can improve food policy development and food environments in recreation and sport facilities [30]. These results are promising because sport venue managers’ perceptions are an important determinant of the success of health promotion interventions [31]. The low-burden, and low-cost of a display-canteen is likely to be attractive to health promotion bodies and have sport industry impact. However, the fact that many managers reported being unable to influence products sold at canteens may limit the scope of the tactic.

We propose that the positive perception and impact of the healthy-canteen display was likely due to many of the perceived barriers being acknowledged and addressed. Firstly, despite the common perception that healthy food is always fresh, and therefore, results in waste and requires refrigeration, the display showed that there are many healthy options which are non-perishable with long shelf life. Secondly, the display showed that healthy items are not always more expensive. Thirdly, the display and pamphlets educated stakeholders on TLL and provided ideas to address barriers related to lack of nutrition knowledge; specifically, the display helped some managers realise that certain foods were healthier than expected (e.g., Chia bars, liquid breakfast drink), and thus there was scope for improvement in their own canteen via easy swaps. Finally, the giftbags provided supply chain information for respondents who did not know where to source healthier options. These findings, along with resources that respondents reported would be helpful (working with the local council, advice on storage/food waste), suggest that there is a need to link community organisations with health promotion bodies that are able to provide resources and support.

Some key reported barriers to stocking healthy options (lack of consumer interest, price and negative financial implications) were not directly addressed by the healthy canteen display. These barriers were also reported in senior community football club canteens in New South Wales [28] and in a New Zealand sport canteen setting [29], and deserve further attention. In relation, our finding that respondents believed stocking healthy foods was important (mean rating 8.5/10) was also reported by Young et al. [11], who found that 66% of NSW football club presidents/secretaries strongly agreed or agreed that clubs should provide more healthy options.

Somewhat paradoxically, although managers believed stocking healthy food was important, they also expressed concerns around consumer interest in healthy foods. However, research has actually shown that the majority of sport organisation members reject ‘junk’ food sponsorship of sporting clubs [9]. Likewise, Australian junior football and American adolescent hockey players and their parents perceive a healthy diet as essential to sporting performance and have an interest in health promotion via sporting clubs [40,41]. Having canteen owners be aware that customers are interested in healthy-canteens may be vital when promoting the adoption of healthy-canteens.

Financial concerns were evident from responses to several questions across this study. In Survey 1, 28% of respondents reported they thought a healthy canteen would reduce sales. In Survey 2, 29% indicated they believed that the ‘price of healthy options was unlikely to be attractive to consumers’ and 27% indicated they believed that swapping to healthier options would have ‘negative financial implication on total sales’. While complete removal of ‘red’ drinks has been shown to reduce revenue at recreational and aquatic centres [21], an RCT of multi-strategic interventions [23] in Australia and a observational study of a TLL intervention in sport and recreational facilities in Canada [18] found that despite changes to purchasing patterns, there was no loss of revenue. Relatedly, a study in Dutch community sport canteens found that increasing the availability and visibility of low-sugar, low-saturated fat, high-protein products, increased sales of these products and total revenue [27]. In future interventions, the importance of replacing ‘red’ products with ‘green’ and ‘orange’ products should be communicated to canteen owners, especially those who are concerned about sales, in order to demonstrate that it is possible to make changes without losing revenue.

A limitation of this study was that there was a relatively small sample and no control group. It is possible external factors (i.e., in addition to viewing the canteen display) were responsible for decisions to make changes to sport canteens. Additionally, a convenience sample was used, thus the initial survey included individuals to whom the intervention might not apply; only 60% of those surveyed had the ability to change their association’s sporting canteen, a likely result of complex venue ownership and management arrangements. There was a high drop-out rate, but it was clear that those who chose to continue to Surveys 2 and 3 were more likely to control purchasing (100%)—while this does introduce some selection bias, it also means the final survey population was more targeted and suited to the aim of the study (influencing change at sporting canteens). The surveys used did not undergo psychometric validation because this is a lengthy process that takes months-years to complete if undertaken properly. Finally, only those canteen operators involved with basketball associations were included. However, it is likely basketball venues share commonalities with other sporting venues and our sample size (22 clubs completing the final survey) was large compared to related studies, which often report on a single community sport canteen.

Future research should evaluate similar interventions, with the inclusion of a comparator group, in other sporting and recreational centres, and consider validating the surveys used. Aspects related to profitability of healthy-canteens and consumers perceptions should be explored further and a better understanding of the community sport canteen environment (ownership and management arrangements) is also required to ensure the right people are targeted to influence changes.

## 5. Conclusions

A healthy-canteen display that educates sport venue managers on TLL and provides supply chain options for healthier food and beverages may help overcome commonly reported barriers to stocking healthy food and drink options at community sport canteens, and encourage sporting venue managers to make changes to their canteens in a self-directed manner. Implementing changes across multiple community sport canteens could improve the nutritional intakes of individuals participating in/observing community sport.

## Figures and Tables

**Figure 1 ijerph-18-10194-f001:**
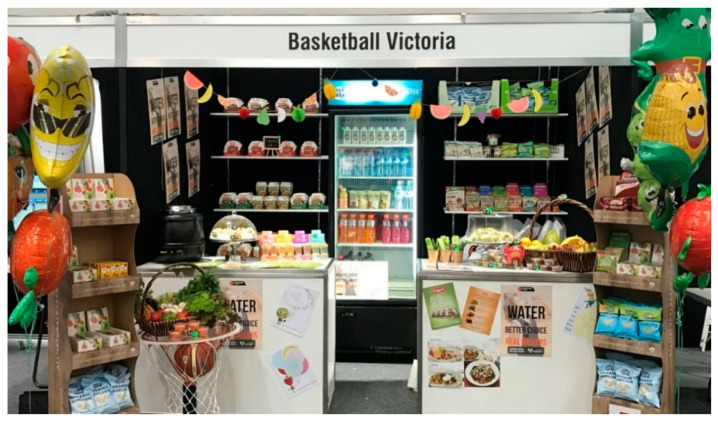
Photograph of visual healthy canteen display set up at Basketball Managers Convention and Trade show, March 2017, Victoria, Melbourne, Australia.

**Figure 2 ijerph-18-10194-f002:**
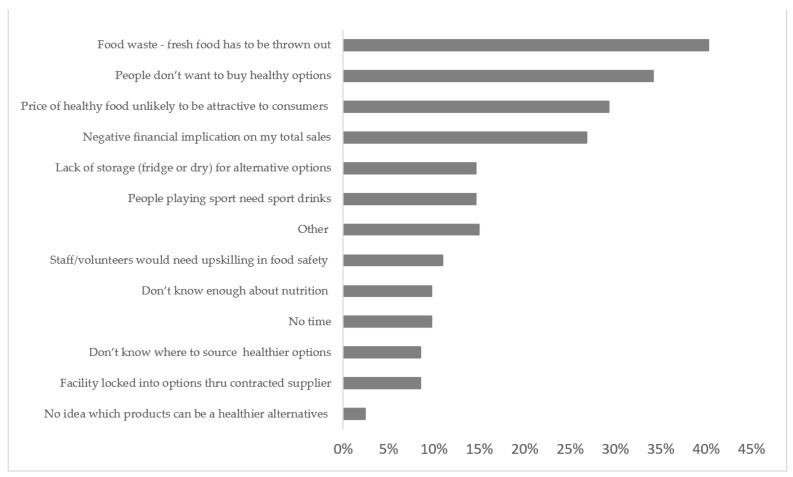
Barriers to stocking healthy options at a community sport canteen.

**Figure 3 ijerph-18-10194-f003:**
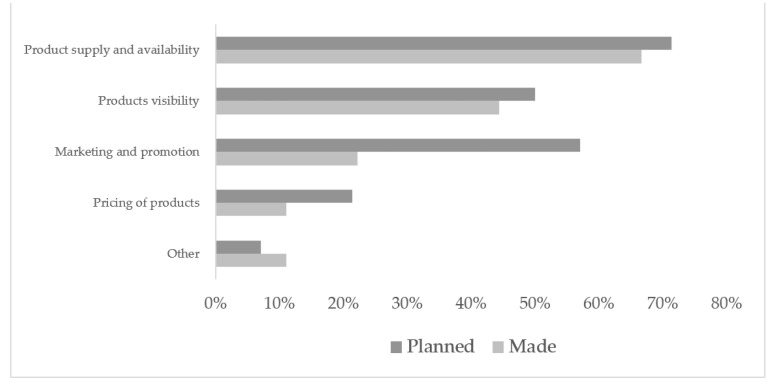
Summary of healthy changes that were made, or have been planned, to sport venue canteens.

**Table 1 ijerph-18-10194-t001:** Canteen information.

Question	1: Baseline (n = 88)	2: Evaluation (n = 76)	3: Follow-Up (n = 22)
Which option best represents your venue canteen: No canteen Privately owned and operated Club owned and operated Council owned and operated Other/missing	11% (n 10) 23% (n 21) 47% (n 42) 13% (n 12) 6 % (n 6)	7% (n 5) 28% (n 21) 54% (n 41) 7% (n 5) 5% (n 4)	5% (n 1) 27% (n 6) 55% (n 12) 14% (n 3) 0% (n 0)
Do you have influence over products available? Yes No	34% (n 30) 66% (n 58)	39% (n 30) 61% (n 46)	68% (n 15) 32% (n 7)

**Table 2 ijerph-18-10194-t002:** Survey 1 (baseline survey) results.

Question	Response
How important do you think it is to have healthy food/drink options at the canteen? (with 0 being not important at all, and 10 being extremely important)	Mean ± SD: 8.5 ± 1.5 Range: 4–10
Do you think that the canteen as it currently operates has appropriate food/beverage options? Yes No	60% (n 53) 40% (n 35)
Have you heard of the food traffic light system before? Yes No	52% (n 46) 48% (n 42)
How confident are you that you could easily distinguish between a “green” healthy, “amber” choose carefully or “red” unhealthy food or drink? (with 0 being not confident at all, and 10 being extremely confident)	Mean ± SD: 7.4 ± 2.2 Range: 2–10
What impact do you think a healthier canteen will have on sales? Increase Reduce No impact	26% (n 23) 28% (n 25) 45% (n 40)
Would you find it useful to see a display canteen which stocks healthier food ? Yes No	92% (n 81) 8% (n 7)

**Table 3 ijerph-18-10194-t003:** Survey 2 (evaluation survey) results.

Question	Response
How useful did you find the healthy eating display canteen in giving you ideas of what healthy options could be sold at your canteen?	Mean ± SD: 8.0 ± 2.2 Range: 0–10
* How important do you think it is to have healthy food/drink options at the canteen? (with 0 being not important at all, and 10 being extremely important)	Mean ± SD: 8.6 ± 1.5 Range: 5–10
* How confident are you that you could easily distinguish between a “green” healthy, “amber” choose carefully or “red” unhealthy food or drink? (with 0 being not confident at all, and 10 being extremely confident)	Mean ± SD: 7.5 ± 2.4 Range: 1–10
Did the healthy eating canteen provide you with any ideas about what your canteen could potentially stock? Yes No	92% (n 70) 8% (n 6)
Did you find the colour coding (green, amber or red) of any of the products surprising? Yes No	50% (n 38) 50% (n 38)
Did the canteen inspire you to make any changes to your canteen? Yes No	75% (n 57) 25% (n 19)

* No statistically significant change from baseline survey.

## Data Availability

The data presented in this study are available on request from the corresponding author.

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
