# Peer review of "Healthy-Canteen Displays: A Tactic to Encourage Community Sport Canteens to Provide Healthier Food and Beverage Options"

_ijerph, 2021, doi:10.3390/ijerph181910194_

Round 1
Reviewer 1 Report
Abstract. Consider changing the first line in the Background to "The community sports setting" or "Community sports settings". As it is written is incorrect grammatically.
Abstract. Consider changing the mean scores (here and throughout the paper) to include only 1 decimal point. You do not have a large enough sample size to measure to the hundredth decimal point.
Abstract/Introduction. Consider defining a 'canteen' -- in the states we call them concession stands. To me a canteen is what a soldier drinks water out of on the battlefield. :-)
Throughout -- be consistent with "sport" vs. "sports" -- "play organized sport" (line 31) vs. "sports participation" (line 35) vs. "via sport" (line 38) vs. "Community sports setting" (line 42), etc. I know I struggle with this -- aim for consistency.
Line 48. I believe you mean "receive funding" and not "receive finding".
Line 99. Here is a good place to define what you mean by "canteen."
Table 2. Consider truncating mean response to one decimal.
Lines 260, 275, 291. Something is wrong with the hard return. The new paragraph starts at the end of the line.
Author Response
Reviewer 1
|
Reviewer comment |
Response |
Changes to manuscript and location |
|
Abstract. Consider changing the first line in the Background to "The community sports setting" or "Community sports settings". As it is written is incorrect grammatically. |
Thank you for picking up on this. This has been edited |
Line 13: Changed to Community sports settings |
|
Abstract. Consider changing the mean scores (here and throughout the paper) to include only 1 decimal point. You do not have a large enough sample size to measure to the hundredth decimal point. |
Thank you for this, we agree and have made these changes throughout. |
Line 23, 382, 388, 408, 417 |
|
Abstract/Introduction. Consider defining a 'canteen' -- in the states we call them concession stands. To me a canteen is what a soldier drinks water out of on the battlefield. :- |
Thank you for this comment, which will ensure our manuscript is understood by readers in North America. We are limited by word count in the abstract, but have provided a full definition in the introduction. In the abstract, we have added (cafeteria) next to the first mention of ‘canteen’ |
Added at Line 17: (cafeteria)
Added at Line 43: Food at community sporting venues is usually supplied via community sports canteens, which are also referred to as stores, kiosks, concession stands or cafeterias. Community sports canteens sell (predominately) readymade and quick-serve food and drink, and are may be run by volunteers. |
|
Throughout -- be consistent with "sport" vs. "sports" -- "play organized sport" (line 31) vs. "sports participation" (line 35) vs. "via sport" (line 38) vs. "Community sports setting" (line 42), etc. I know I struggle with this -- aim for consistency |
Thank you for picking up on this. In line with IJEPRH use of American English, we have gone with ‘sport’ throughout. |
Total of 41 changes made throughout manuscript. |
|
Line 48. I believe you mean "receive funding" and not "receive finding" |
Yes, we do. This typographical errors has been corrected. |
Line 93 ‘Finding’ changed to ‘Funding’ |
|
Line 99. Here is a good place to define what you mean by "canteen." |
Thank you, a definition and additional context around canteens has been added. |
Added at Line 46: Food at community sporting venues is usually supplied via community sport canteens, which are also referred to as stores, kiosks, concession stands or cafeterias. Community sport canteens sell (predominately) readymade food and drink, and are usually run by volunteers. |
|
Table 2. Consider truncating mean response to one decimal. |
Responses now reported to 1dp. |
Line 208 |
|
Lines 260, 275, 291. Something is wrong with the hard return. The new paragraph starts at the end of the line. |
Formatting corrected. |
Throughout |
Reviewer 2 Report
The paper aims to evaluate perceptions, knowledge, and impact of a healthy-canteen display, based on traffic light labeling (TLL). The topic fits within the scope of the Journal and is interesting for a potential reader.
Please allow me some comments on the manuscript:
The abstract (page 1, lines 13-27) is mostly clear. However, I consider there are two missing pieces of information that should appear: the aim of the study and setting (Australia or similar). This would help the reader to fully understand the paper.
Page 1, line 32 (among others). As far as I have been informed, the in-text citations do not follow the Journal guidelines. Please carefully check the Journal guidelines and correct them, if applicable.
The Materials and Methods section (pages 2 and 3) is the one that concerns me the most:
- A better explanation of the study design is required (for example, the division into three surveys).
- I consider the number of participants should appear in the 2.1 Study Design, Setting and Participants section.
- A full description of the characteristics of the sample should be added, including sociodemographic data, among others.
Therefore, a deeper explanation of the study design, setting, and participants is compulsory. The description should allow other researchers to fully understand even replicate your study.
Page 3, line 125. The authors state that t-test was applied. However, I cannot find the t-test nor significant differences throughout the paper. I believe this aspect needs clarification.
Table 2 (page 5) includes the mean and range in the “Response” column. I consider this is appropriate, but Standard Deviation (SD) would also be useful.
Minor comments
I consider the Figures' title/description (for example, after Figure 2), is too long. I would like to invite the authors to consider a shorter title and to include the rest of the information in the main text of the article.
Page 2, lines 52 and 84: Please check the spelling of “LABELING”.
I hope the comments are useful.
Author Response
|
Reviewer comment |
Response |
Changes to manuscript and location |
|
The paper aims to evaluate perceptions, knowledge, and impact of a healthy-canteen display, based on traffic light labeling (TLL). The topic fits within the scope of the Journal and is interesting for a potential reader.
|
Thank you for this comment. |
NA |
|
The abstract (page 1, lines 13-27) is mostly clear. However, I consider there are two missing pieces of information that should appear: the aim of the study and setting (Australia or similar). This would help the reader to fully understand the paper. |
Thank you for these comments. We have now included an aim and mention of Australia. |
Added at Line 16: The aim of this project was to evaluate perceptions, knowledge and the impact (e.g. barriers and outcomes) of a healthy-canteen display, based on traffic light labelling (TLL), that was set up at an Australian Basketball Association Managers’ Convention and Trade Show.
|
|
Page 1, line 32 (among others). As far as I have been informed, the in-text citations do not follow the Journal guidelines. Please carefully check the Journal guidelines and correct them, if applicable. |
Thank you. All citations have now be changed to square brackets and no sub/superscript. |
Throughout manuscript. |
|
The Materials and Methods section (pages 2 and 3) is the one that concerns me the most: A better explanation of the study design is required (for example, the division into three surveys). AND Therefore, a deeper explanation of the study design, setting, and participants is compulsory. The description should allow other researchers to fully understand even replicate your study. |
Thank you for this comment. The study design involved (1) Set up of a canteen display at a convention and (2) Surveying of conference delegates, before viewing, directly after viewing and three months after visiting the stand. We have amended sub-headings to Study design, Setting, Participants, Healthy canteen set-up, Healthy canteen evaluation, Survey details, and Ethical Considerations. We feel these headings will help the reader understand the materials and methods.
Setting: We already described the setting as “a Basketball Association Managers’ Convention and Trade show, 9 to 11 March 2017, Melbourne, Victoria, Australia”. For clarity, we have added an additional statement on the convention to the manuscript.
Participants: We already explained that “All delegates aged 18 years and over who attended the convention were invited to participate”. We now explicitly state that exclusion criteria were not applied. Sociodemographic data was not collected (please refer comment below for justification and explanation).
Design/Canteen display: We have already explained that the canteen was set up based on TLL (green, red amber products) and provided a picture of the display canteen. In order to ensure other researchers can replicate this, we have now added further explanation on TLL and references to assist with classifying items as green, amber and red. We believe this is enough information for other researchers to set up a similar display.
Design/division into three surveys: The division into three surveys was already explained as follows:
“The impact of the display canteen on managers’ perceptions of healthy-canteens and TLL was assessed via a short questionnaire, administered using an iPad, before delegates visited the stand (survey 1: baseline survey) and immediately after they visited the stand (survey 2: evaluation survey). All participants were approached on entry to the convention to complete survey 1 and all participants who visited the display were invited to complete survey 2. After completing survey 2, we gave respondents a gift bag with samples of healthy packaged food, lists of suppliers, costing ideas, and suggestions on how to make changes to their canteens. The potential of the display and gift bags to encourage managers to stock healthier foods was assessed via short survey (survey 3: follow-up survey), administered via email to all participants who had completed survey 2, three months after the convention”
Due to word limit constraints, we originally presented detailed information on each survey as supplementary material. However, for clarity, we have moved this information to the main text. Due to this edit, the manuscript now exceeds the 3000 word limit. |
Added at Line 260: The convention focused on “facility management, programs and activities, technology, human resources and high performance”.
Added line 265: and no exclusion criteria were applied
Added line 273: Green, amber, and red classifications were based on pre-defined criteria for saturated fat, added sugar, and/or salt, energy and fiber quantity[18]. Examples of green, amber, red foods include fresh fruit, muesli bars, and chocolate respectively. For further details on how we classified items, we direct readers to [33], [34] and [35]
Added line 305: Variables and data sources: All surveys (1-3) asked who ran and operated the club canteen (no canteen/private/club/council/other) and if respondents could influence products stocked at the canteen (yes/no). Survey 1, 2, and 3 had an additional nine, eight and five questions, respectively. Response options for surveys 1, 2 were yes/no, Likert-scales (1-10) or open-ended. Response options for survey 3 were yes/no or multiple choice. Survey 1 (baseline survey): Two questions assessed awareness of, and confidence in implementing the traffic light system. The remaining questions assessed perceived: healthfulness of current foods stocked at the canteen; importance of having healthy canteen options; barriers to implementing a healthy canteen; impact of healthy canteens on sales; and utility of seeing a display healthy canteen. The open-ended questions probed respondents about currently available healthy canteen options and what they thought a healthy canteen would look like. Authors derived barriers to implementing a healthy canteen based on existing literature[29]. Survey 2 (evaluation survey): Two questions on perceptions of importance of including healthy foods and confidence in using the traffic light system were directly repeated from survey 1. The remaining four questions assessed how useful delegates found the canteen, whether the display provided ideas for products to stock, if the traffic light system was surprising, and whether the canteen inspired delegates to make any changes to their canteen. The open-ended questions asked: what ideas respondents liked, what surprised them with regards to traffic light classifications, what other information they would find helpful, and what types of changes they would consider making. Survey 3 (follow-up survey): The follow-up survey checked if respondents visited the healthy canteen display at the Basketball Managers Convention and Trade Show. The remaining items asked if any changes had been made and what these were, or if any changes were planned and what these were. Due to time constraints, the surveys were not validated. |
|
I consider the number of participants should appear in the 2.1 Study Design, Setting and Participants section. |
In this section we already reported ‘A total of 164 associations were invited to attend but the actual number of attendees on the day was not recorded’. We felt describing the number of participants flowed better in the results, once each of the surveys have been introduced. However, for clarity have also added the number of participants undertaking survey 1 to the methods section. |
Added Line 267: 88 delegates participated in the current study. |
|
A full description of the characteristics of the sample should be added, including sociodemographic data, among others. |
Thank you for this suggestion. We did not collect sociodemographic data as the aim was to survey all attendees at the basketball conference in a simple manner that maintained their anonymity. We have explained that all delegates aged 18 years over who attended the basketball conference were included in the study and have added information to further clarify why sample characteristics are not reported. |
Added Line 364: In order to ensure anonymity, and in line with our planned data analyses, socio-demographic data was not collected. |
|
Page 3, line 125. The authors state that t-test was applied. However, I cannot find the t-test nor significant differences throughout the paper. I believe this aspect needs clarification. |
We had stated “Viewing the display did not lead to a statistically significant change in mean rating for importance of stocking healthy and confidence in classifying foods according to TLL” (deleted from line 435) We have now added specific t-test statistics, as well as adding a new table (Table 3) outlining survey 2 results. |
Added Line 228: Viewing the display did not lead to a statistically significant change in mean rating for importance of stocking healthy options, t (-0.825), df =162, p=0.410. Likewise, there was no statistically significant change in confidence in classifying foods according to TLL be-tween survey 1 and survey 2, t(-0.558), df=162, p=0.578 (Table 3).
Added Line 410: Table 3 |
|
Table 2 (page 5) includes the mean and range in the “Response” column. I consider this is appropriate, but Standard Deviation (SD) would also be useful. |
SD has now been added to tables |
Line 214, 231 |
|
I consider the Figures' title/description (for example, after Figure 2), is too long. I would like to invite the authors to consider a shorter title and to include the rest of the information in the main text of the article. |
Thank you for this suggestion. The figure/table legends have been truncated. No additional information was added to the main text, as this had already been explained when describing the surveys (Line 319, 333) . |
Deleted from Figure 2 (Line 404): Basketball venue/club managers answered this question before viewing a healthy-canteen display at a Basketball Association Managers’ Convention and Trade Show. They were instructed to choose all options that applied.
Deleted From Figure 3 (Line 457): Basketball venue/club managers answered this question as part of a follow-up survey, three months after viewing a healthy-canteen display at a Basketball Managers’ Association Convention. They were instructed to choose all options that applied.
|
|
Page 2, lines 52 and 84: Please check the spelling of “LABELING”. |
British/Australian spelling is two “L’s”. American is one. We have edited to match American spelling, in line with the journal’s preferences. |
Line 17, 97, 29 |
|
I hope the comments are useful. |
These have been very helpful and we believe the feedback has strengthened our manuscript. Thank you. |
NA |
Reviewer 3 Report
It is an interesting topic and I congratulate the authors on the research, but indicate some comments to improve the manuscript.
Introduction section:
- It is well planned,
- Add the research aim at the end of this section.
Materials and methods section
- The information must be expanded and must contain the following sections for separate: design / type of study (ok), sample / participants / selection criteria (must be completed), study variables (must be completed), procedure / intervention (more information is needed), statistical analysis (ok ) and ethical considerations (must be completed).
Results section
- It is well planned, pero parte de este apartado debe utilizarse para la sección de material y methods
Discussion section:
- The first paragraph of the discussion should indicate a summary of the main findings of the study.
- The third paragraph should contain bibliographic references and discuss the information with other research.
Author Response
|
Reviewer comment |
Response |
Changes to manuscript and location |
|
It is an interesting topic and I congratulate the authors on the research, but indicate some comments to improve the manuscript.
|
Thank you for these encouraging comments. |
NA |
|
Introduction section: It is well planned, Add the research aim at the end of this section. |
Thank you for this comment. The introduction does already describe the aim and hypotheses as follows; Accordingly, the aim of this project was to evaluate perceptions, knowledge and the impact (e.g. barriers and outcomes) of a healthy-canteen display, based on traffic light labelling (TLL), that was set up at a Basketball Association Managers’ Convention and Trade Show. We hypothesised that the intervention would positively impact managers’ perceptions of healthy-canteens and their ability to introduce healthier items in their local contexts. This project was supported by the Victorian Health Promotion Foundation (VicHealth), the broad aims of which are to create healthier sport environments, increase access to drinking water and make healthier food and beverage choices the easy choice in a range of environments. |
NA |
|
Materials and methods section The information must be expanded and must contain the following sections for separate: design /type of study (ok), sample / participants / selection criteria (must be completed), study variables(must be completed), procedure / intervention (more information is needed), statistical analysis(ok ) and ethical considerations (must be completed). |
Thank you. We were limited by the word count when describing the methods but have now expanded these, as per the reviewers’ suggestions.
New materials and methods sub-headings are: Study design, settings, participants, healthy canteen display set up, healthy canteen display evaluation, survey details, statistical analyses, ethics considerations
sample / participants / selection criteria We explained that All delegates aged 18 years and over who attended the convention were eligible to participate (convenience sample). A total of 164 associations were invited to attend but the actual number of attendees on the day was not recorded. We have now also specified that no exclusion criteria were applied, expanded on the purpose of the convention where the study took place, and explained why further sociodemographic data were not collected/reported.
procedure / intervention (more information is needed) We have added additional detail on the set up of the canteen as well as moving the description of the study questionnaires to the main text (section 2.6 survey details)
Ethical consideration We have added a statement on this as section 2.8 to the manuscript. |
Added at Line 108: The convention focused on “facility management, programs and activities, technology, human resources and high performance”.
Added line 113: and no exclusion criteria were applied
Added line 121: Green, amber, and red classifications were based on pre-defined criteria for saturated fat, added sugar, and/or salt, energy and fiber quantity[18]. Examples of green, amber, red foods include fresh fruit, muesli bars, and chocolate respectively. For further details on how we classified items, we direct readers to [33], [34] and [35]
Added line 143: Variables and data sources: All surveys (1-3) asked who ran and operated the club canteen (no canteen/private/club/council/other) and if respondents could influence products stocked at the canteen (yes/no). Survey 1, 2, and 3 had an additional nine, eight and five questions, respectively. Response options for surveys 1, 2 were yes/no, Likert-scales (1-10) or open-ended. Response options for survey 3 were yes/no or multiple choice. Survey 1 (baseline survey): Two questions assessed awareness of, and confidence in implementing the traffic light system. The remaining questions assessed perceived: healthfulness of current foods stocked at the canteen; importance of having healthy canteen options; barriers to implementing a healthy canteen; impact of healthy canteens on sales; and utility of seeing a display healthy canteen. The open-ended questions probed respondents about currently available healthy canteen options and what they thought a healthy canteen would look like. Authors derived barriers to implementing a healthy canteen based on existing literature[29]. Survey 2 (evaluation survey): Two questions on perceptions of importance of including healthy foods and confidence in using the traffic light system were directly repeated from survey 1. The remaining four questions assessed how useful delegates found the canteen, whether the display provided ideas for products to stock, if the traffic light system was surprising, and whether the canteen inspired delegates to make any changes to their canteen. The open-ended questions asked: what ideas respondents liked, what surprised them with regards to traffic light classifications, what other information they would find helpful, and what types of changes they would consider making. Survey 3 (follow-up survey): The follow-up survey checked if respondents visited the healthy canteen display at the Basketball Managers Convention and Trade Show. The remaining items asked if any changes had been made and what these were, or if any changes were planned and what these were. Due to time constraints, the surveys were not validated.
Added line 182: The front page of each online survey included background information about the research, along with participant ethical considerations should they choose to complete it. Given the low-risk nature of the research, consent to take part was implied through the lack of objection to complete the relevant survey (see Institutional Review Board Statement for ethics approval details).
Added line 190: In order to ensure anonymity, and in line with our planned data analyses, sociodemographic data was not collected |
|
Results section It is well planned |
Thank you. |
NA |
|
Discussion section: The first paragraph of the discussion should indicate a summary of the main findings of the study. |
We agree, and have already provided a summary of key results as the closing lines of the first paragraph of the discussion, as follows: The key findings were: (1) basketball managers, like presidents in related sport settings[28], believed stocking healthy food was important but reported barriers to having a healthy-canteen; (2) over 90% of basketball managers found the healthy-canteen display helpful; (3) most attendees were inspired to make changes to their own canteens after viewing the healthy-canteen display; and (4) over 40% of respondents to the final follow-up survey reported making changes to their canteen. |
NA |
|
The third paragraph should contain bibliographic references and discuss the information with other research. |
Thank you for this suggestion. The aim of the 3rd paragraph is to provide our own perception of the study findings, and thus we do not feel comparison to other studies is required here.
We have now added a brief statement to discuss how information around belief in stocking healthy foods compares to other research. We had also already provided bibliographic references and discussed the information with reference to other research throughout the remainder of the discussion e.g. - This is consistent with existing literature showing that organisational capacity building can improve food policy development and food environments in recreation and sport facilities.[30] - These barriers were also reported in senior community football club canteens in New South Wales[28] and in a New Zealand sport canteen setting[29]
Any additional direct comparisons with existing literature are limited due to the novelty of our study.
|
Added Line 317: In relation, our finding that respondents believed stocking healthy foods was important (mean rating 8.5/10) was also reported by Young et al. (11), who found that 66% of NSW football club presidents/secretaries strongly agreed or agreed that clubs should provide more healthy options. |
Round 2
Reviewer 2 Report
The authors have nicely addressed point by point all the concerns raised in the previous round. Now I feel the paper is more consistent and has more relevance for the potential reader. Congratulations on a good job